# Investigational Microbiological Therapy for Glioma

**DOI:** 10.3390/cancers14235977

**Published:** 2022-12-02

**Authors:** Jing Wang, Yaxue Liu, Aohan Zhang, Wenxin Yu, Qian Lei, Bo Xiao, Zhaohui Luo

**Affiliations:** 1Department of Neurology, Xiangya Hospital, Central South University, Changsha 410008, China; 2Xiangya School of Medicine, Central South University, Changsha 410083, China; 3National Clinical Research Center for Geriatric Disorders, Xiangya Hospital, Central South University, Changsha 410008, China; 4Clinical Research Center for Epileptic Disease of Hunan Province, Central South University, Changsha 410008, China

**Keywords:** microorganism, targeted therapy, gut–brain axis, intestinal flora, glioma treatment

## Abstract

**Simple Summary:**

Glioma is a life-threatening malignancy where conventional therapies are ineffective, and microorganisms are a promising weapon that can be used in cancer treatment. In order to promote the use of microbial therapies in glioma, this article summarizes the microorganisms that have been used for glioma treatment in recent years and their mechanisms and how the gut flora affect glioma progression, suggesting the current limitations of this approach and possible future directions for its development. This paper may bring new inspiration to those who investigate glioma, promote the progress of glioma therapy research, and bring new promise to glioma patients.

**Abstract:**

Glioma is the most common primary malignancy of the central nervous system (CNS), and 50% of patients present with glioblastoma (GBM), which is the most aggressive type. Currently, the most popular therapies are progressive chemotherapy and treatment with temozolomide (TMZ), but the median survival of glioma patients is still low as a result of the emergence of drug resistance, so we urgently need to find new therapies. A growing number of studies have shown that the diversity, bioactivity, and manipulability of microorganisms make microbial therapy a promising approach for cancer treatment. However, the many studies on the research progress of microorganisms and their derivatives in the development and treatment of glioma are scattered, and nobody has yet provided a comprehensive summary of them. Therefore, in this paper, we review the research progress of microorganisms and their derivatives in the development and treatment of glioma and conclude that it is possible to treat glioma by exogenous microbial therapies and targeting the gut–brain axis. In this article, we discuss the prospects and pressing issues relating to these therapies with the aim of providing new ideas for the treatment of glioma.

## 1. Introduction

Glioma is the most common primary malignancy of the CNS. The annual occurrence rate of glioma is 3–6.4/100,000, accounting for about 23.3% of all central nervous system tumors and about 78.3% of malignant tumors [1]. They usually originate from glial cells or precursor cells and develop into astrocytomas, oligodendrogliomas, ventricular meningiomas, or oligodendrocytomas [2]. According to the World Health Organization classification, gliomas are classified into four grades, with relatively high grades being associated with a poorer prognosis. GBM is the most common and most malignant type of grade 4 glioma [3]. It has surpassed pancreatic cancer and hepatocellular carcinoma to become the most intractable tumor [4]. Patients with glioma commonly experience symptoms and signs including seizures, focal neurological deficits, and headaches. It is difficult to cure and is prone to recurrence, placing a huge burden on society. To date, conventional treatment for glioma includes surgical resection, TMZ, and radiation therapy, which is not enough to fight the cancer, and the treatments have many disadvantages. Conventional therapies tend to cause pharmacological side effects in normal cells, lack the ability to penetrate solid tumor tissue, and cannot eradicate all cancer cells in gliomas because of drug resistance. Thus, there is an immediate need to develop new therapies that can complement or replace conventional therapies used to treat cancer. The use of bacteria for cancer treatment is a unique therapeutic option in this regard.

The microbiota and human cancer have been closely linked throughout history. The role of microorganisms in cancer has been controversial for centuries. More and more studies have shown that microorganisms not only are involved in tumor formation [5] but also can be used to treat tumors. As a type of prokaryotic microorganism, bacteria have great potential for use in cancer treatment. They can be genetically manipulated to become nonpathogenic, and their unique virulence factor can be used as a weapon against tumors. They can proliferate in tissues and can be controlled in numbers by antibiotics, making bacteria viable in vivo microdrugs for cancer treatment [6]. In recent years, much experimental evidence has shown that there are bidirectional communication channels between the gut and the brain, involving neural, endocrine, and inflammatory mechanisms. Communication through these channels can be regulated by changes in the permeability of the intestinal wall and the blood–brain barrier (BBB) [7]. There is also an interaction between intestinal flora and glioma. In this article, we summarize the impact of microorganisms and their derivatives on the development and treatment of glioma from two aspects: the use of exogenous microorganisms and their derivatives for the treatment of glioma and the impact of the patient’s own gut microorganisms on the glioma microenvironment. In this article, we discuss the prospects and pressing issues of these therapies. It is our hope that they will provide new ideas for the treatment of glioma.

## 2. Microorganisms Influence Glioma Development

The vast majority of cancers are associated with environmental factors, including physical, chemical, and biological factors. Among these biological factors are bacterial and viral infections. Bacteria can directly manipulate their host cells at various stages of their infection cycle, and this manipulation can affect the integrity of the host cells and may lead to cancer formation [8]. For example, some bacterial toxins can break double-stranded DNA in host cells, making them susceptible to DNA mutations and deletions during the process of repairing, which can lead to cancer [9]. There are also certain bacterial toxins that promote carcinogenic effects by inducing resistance to cell death signaling and by promoting proliferation signaling [10]. In addition, bacterially induced inflammation may also be a major causative agent of tumors. On the one hand, inflammation promotes cellular chemotaxis and increases the chances of cellular carcinogenesis, and on the other hand, inflammation triggers immunosuppression and provides a suitable background for the development of tumors [11]. Some epidemiological evidence suggests that the presence of zoonotic viruses or bacteria in the feces of domestic animals may be a possible cause of glioma [12].

The topics of viral infections and glioma have been controversial for a long time. Current studies focus on cytomegalovirus, polyomavirus, adenovirus, herpesvirus, etc. A meta-analysis has shown an association between human cytomegalovirus (HCMV) and glioma [13], and the mechanism may be related to the stress response of glioma cells triggered by HCMV through the downregulation of specific miRNAs [14]. Polyomavirus and adenovirus infections are also associated with glioma development [15], and adenovirus infection promotes GBM stem cell formation through the TLR9/NEAT1/STAT3 pathway [16]. Similar to bacterial oncogenesis, viral oncogenesis also regulates proliferation, anti-apoptosis, and immune escape activities through the interaction of specific viral gene products with cellular gene targets [17]. Although the mechanisms of oncogenic viruses in glioma are not well understood, it is interesting to note that these “oncogenic” viruses can also play a role in “curing” cancer with appropriate modifications.

## 3. Exogenous Microorganisms and Their Derivatives for Glioma Treatment

Glioma is a life-threatening malignancy, and traditional radiotherapy and chemotherapy are not very effective. A growing number of studies have shown that microorganisms and their derivatives can be used as cancer therapies. We summarize current research advances in three respects: microbial derivatives for glioma treatment; tumor lysis by microorganisms; and microbial targeting of gliomas for drug delivery. Some of the research is compiled in Table 1.

### 3.1. Derivatives of Microorganisms Are Used to Treat Gliomas

Bacteria can produce a range of bioactive substances that are used in the treatment of gliomas through different mechanisms, such as bacteriocins, bacterial toxins, and enzymes.

Bacteria use ribosomes to synthesize bacteriocins and antimicrobial peptides that prevent other bacterial strains from invading their own ecological niches. Bacterial peptides can also prevent the growth of tumor cells. Prodigiosin causes cell death by activating the JNK pathway and reducing the AKT/mTOR pathway in GBM cells [18]. As a PAMP, flagellin is recognized by TLR and activates the body’s immune system. Flagellin can reduce the number of myeloid-derived suppressor cells (MDSCs) in tumor tissues and adjust the conversion of tumor-associated macrophages (TAMs) from M2 to M1 type. Therefore, it promotes the immune response in the glioma microenvironment and inhibits tumor growth [19]. Someone combined them and found that they significantly reduced the size of intracranial tumors in mice. P53 is a proapoptotic transcription factor. TSA leads to p53 phosphorylation by activating p38 mitogen-activated protein kinase (p38MAPK). TSA inhibits NF-κB activity by inducing IKK dephosphorylation [21]. TSA also inhibits GBM vascular proliferation [32]. MOX is able to induce cell apoptosis by increasing the Bcl-2-associated X protein/B-cell lymphoma 2 ratio and activating the caspase-3/-9 cascade in glioma, and it induces G0/G1 cell cycle arrest and apoptosis to inhibit the viability of glioma cells [22].

Bacteria express and release specific toxins, which are a class of highly poisonous proteins generated and released by bacteria with specific functions. Due to their great toxicity, bacterial toxins have been demonstrated to be effective cancer treatments. PE, the most aggressive virulence factor produced by Pseudomonas aeruginosa, can be employed to destroy tumor cells by inhibiting protein synthesis via ADP-ribosylation of eukaryotic elongation factor 2. PE can be combined with antibodies or receptor ligands for precise tumor targeting to generate chimeric proteins, and this toxin-linked targeting element forms an immunotoxin that can be exploited for cancer therapy [33]. The targeting element is responsible for attaching tumor surface molecules, while the toxin kills cancer cells. Vibrio cholerae produces cholera toxin, cholera toxin subunit B (CTB), which facilitates glioma-targeted drug delivery by targeting sphingolipid GM1 expressed in the BBB, neovascularization, and glioma cells [24]. *E. coli* produces CNF1, which causes glioma cells to overexpress p21 and p16, promotes glioma cell senescence, blocks tumor progression and migration, and protects the structure and function of healthy surrounding tissues [25]. CTX, a peptide derived from scorpion venom, penetrates the BBB and preferentially recognizes and targets glioma cells; therefore, a group of researchers developed a chimeric protein of CTX-CNF1 that dramatically prolongs the survival time of glioma mice following systemic injection [26].

Essential amino acids are necessary for cell growth and cellular metabolism, and their depletion is one of the cancer treatments. Arginine deiminase (ADI) derived from Streptococcus pyogenes depletes arginine from tumor cells, resulting in a nutritional deficiency and inhibiting tumor cell development. The addition of suberoylanilide hydroxamic acid (SAHA) to ADI can further increase the strong tumorolytic effect [27]. Recent studies have demonstrated that the therapeutic effect of Arg deprivation is largely dependent on the expression of argininosuccinate synthase in tumors, regardless of whether they are of the nutrition-deficient type [34], but this therapy is not hindered by the BBB, provides multiple mechanisms for inducing apoptosis in glioma cells, and has enormous research potential.

To date, temozolomide is the only first-line chemotherapeutic agent for high-grade glioma, but drug resistance in tumors is also a significant problem, and the development of new drugs is a focus of current research. The derivatives of these microorganisms mentioned above have achieved a good effect against glioma in animal experiments and cellular experiments, so we believe that these substances may be helpful for the development of new drugs. Thousands of microorganisms cause problems in humans while potentially contributing to drug development (Figure 1).

### 3.2. Microbial-Targeted Therapy for Glioma

#### 3.2.1. Potential of Bacteria

William Coley, an American surgeon, administered Streptococcus pyogenes to cancer patients in 1891, marking the first use of bacteria to treat cancer. As medicine has evolved, researchers have discovered many types of bacteria that can be used in tumor treatment. In theory, bacterial therapies have many advantages over conventional treatments, such as the ability to target tumors and actively penetrate tumor tissue, which may be related to the unique hypoxic microenvironment of tumor tissue and the immunosuppression caused by tumors [35,36]. Bacteria can, on the one hand, kill tumors through their own toxicity and, on the other hand, activate immune cells in the tumor microenvironment, enhancing the clearance of tumors by the immune system [37]. The BBB consists of several components, including tight junctions, vascular basement membranes, and astrocyte terminals that cover endothelial cells to form a physical barrier that functions as a filtering barrier for capillaries. As a filtering barrier, the BBB prevents most anticancer agents from penetrating the tumor and limits the therapeutic effect, which is one of the reasons for the poor prognosis of glioma [38]. Many bacteria can cross the blood–brain barrier and enter the center through a unique mechanism, which sets the stage for bacterial entry into targeted gliomas [39]. Bacterial therapy has been extensively studied in tumors of different tissues [40]. Theoretically, it seems that this therapy could also be used for gliomas, but there are still few studies on gliomas. The brain is more delicate than other tissues, and bacterial infection will have serious consequences. In the event of a brain abscess and brain hemorrhage, patients will be at risk of death [41]. How can the damage caused by bacteria to nontumor tissues be mitigated? How can the occurrence of bacteremia and shock be avoided? How can we improve the efficiency of tumor treatment? How can we enhance the targeting of bacteria to tumors? Gene-editing technology may be the solution to these problems.

How might genetically modified microorganisms be utilized to improve treatment results? We propose that we can begin with the following four factors: First, to increase safety and lower virulence without eliminating anticancer action, it is frequently necessary to remove the key virulence genes of recognized pathogens in order to reduce their pathogenicity. For instance, phoP and phoQ are both virulence genes of Salmonella typhi, and deletion of these two genes decreases the pathogenicity of the bacteria while leaving the tumor-killing characteristics mostly unaffected [42]. We can make nutrient-deficient mutants, and the tumor may manufacture chemicals that allow these mutants to thrive solely within the tumor. Second, to enhance tumor targeting, bacteria can be modified to express tumor-specific binding proteins on their surface, therefore increasing the affinity between the bacteria and the tumor. Bacteria can also be guided by external energy, such as laser stimulation, ultrasound, and magnetic waves; ultrasound-controlled bacteria have been employed in cancer immunotherapy to direct bacterial migration toward tumors [43]. Third, the tumor microenvironment (TME) can be modulated by bacteria, for instance, bacteria can stimulate the immune system and inhibit tumor angiogenesis by releasing cytokines and immune checkpoint inhibitors, and some bacteria can steal nutrients, and some photosynthetic bacteria produce oxygen while ameliorating the hypoxic condition of the tumor microenvironment. Finally, gene-editing tools will allow us to adjust drug expression strategies. Hypoxia, acidosis, and necrosis are TME characteristics that bacteria can detect and employ to improve tumor specificity; thus, drug expression can be connected to these characteristics. For instance, acidic pH can activate particular promoters in Salmonella [44]; thus, inserting drug-encoding genes downstream of them can increase the likelihood of drug release in the TME. Overall, the complexity of bacteria as living organisms influences the difficulty and risk of transforming them into anticancer weapons, but it also enables scientists to fine-tune the unique roles of different strains to achieve antitumor activity that is not feasible with traditional therapy.

#### 3.2.2. Oncolytic Virus

The concept of an oncolytic virus (OV) was first demonstrated in a case report in 1912 when DePace described a woman with cervical cancer who exhibited tumor regression after receiving an attenuated rabies virus vaccine [45]. With the development of viral genetic engineering techniques, the potential of lysing viruses was discovered. OVs have been more extensively studied in glioma than the bacteria mentioned above. Indeed, there are some features that make gliomas particularly suitable for lytic virus therapy. Gliomas are confined to the brain and lack distant metastasis, which favors the intratumor spread of OVs. The tumor growth is mainly surrounded by postmitotic cells, which facilitates virus replication with an active cell cycle [46].

There are multiple reasons for OVs targeting glioma. Some viruses have their own mechanisms or use immune cells as carriers that can cross the BBB to reach the site of glioma when administered systemically [47]. Receptors for some viruses can be highly expressed in tumor cells, such as the Poliovirus receptor CD155-targeted oncolysis of glioma [48]. Tumor tissue often exhibits viral immune defects. In normal tissues, the interferon pathway activates the downstream cascade signaling and activates immune cells, while tumor tissues are interferon-deficient [49]. Protein kinase R (PKR) is a strong inhibitor of viral protein synthesis and tumors, with activated Ras pathways showing impaired PKR function [46]. Tumor cells are replicatively and metabolically active, which also facilitates the replication of the virus.

Similarly, OVs not only have a direct tumorolytic effect on cancer cells but also boost the immune system’s antitumor response. When infected cancer cells are lysed, they release additional infectious viral particles that aid in the destruction of the remaining tumor. OVs can induce glioblastoma autophagy and also break the immunosuppressed glioma microenvironment, activate the immune system, positively modulate immune synapses, and block immunosuppressive tumor metabolic circuits to inhibit tumor growth in different ways [50,51,52].

Viruses, such as herpes simplex virus-1, adenovirus, vaccinia virus, myxoma virus, and parvovirus, have been considered as glioma lysis agents [53]. Herpes simplex virus type 1 (HSV-1) is an enveloped double-stranded DNA virus known to infect and replicate in neural tissue, making it a potential treatment for glioma. Utilizing gene-editing techniques has resulted in a HSV with enhanced properties. γ34.5 is a viral antagonistic protein known to block protein kinase R (PKR) antiviral signaling in infected cells. HSV-1716 was obtained by deleting two copies of γ34.5/RL1 with weak toxicity. HSV-1 G207 is made safer by the insertion of the *Escherichia coli* lacZ gene into the coding sequence for the viral ICP6 gene and deletion of both copies of γ34.5 loci within the viral genome [53]. HSV-1716 and HSV-1 G207 have both been utilized in clinical research [54]. ICP47 supports HSV1 proliferation by reducing host cell-induced immune destruction. It was obtained by deleting the ICP34.5 and ICP47 from the appropriate locations to obtain talimogene laherparepvec, sold under the brand name Imlygic (T-VEC). T-VEC has passed phase III clinical trials and has been approved for clinical use by the FDA and the European Medicines Agency [54]. PVSRIPO, a chimera composed of poliovirus and rhinovirus, also completed the phase I clinical trial and received breakthrough treatment. Poliovirus has an affinity for CD155, which is highly expressed in gliomas and mediates its targeting to tumors. Intratumoral infusion of PVSRIPO causes the activation of antiviral responses and increases immune system activity, which provides an ideal platform for antitumor immunity [55]. Proinflammatory chemokines and neutrophils were increased in the tumor microenvironment treated with PVSRIPO. Neutrophils are able to produce TNF-α, induce nitric oxide synthase (iNOS), and also regulate the functions of NK, T, and B cells, which have long-lasting antitumor effects [56]. Parvovirus H-1 (H-1PV) is an oncolytic single-stranded DNA virus. The natural hosts of H-1PV are rodents, which have the ability to infect and replicate in humans, but lack pathogenicity. H-1PV interacts with galectin-1 on the cell surface and uses this glycoprotein to enter cancer cells [57], and its tumorolytic mechanism of action is thought to work through the cathepsin-mediated cell death pathway, so it may be an effective therapeutic approach for targeting glioma cells with defects in the apoptotic pathway [58]. It is able to cross the blood–brain barrier and therefore has potential for intravenous administration [59]. In phases I and II clinical studies, it met the primary objectives of safety and tolerability. There were no signs of systemic inflammation, excessive immune activation, or main organ toxicity [60]. In short, it is also a promising virus for targeting glioma.

Treating gliomas with oncolytic bacteria also presents challenges, such as side effects. Common adverse effects include cerebral edema, hemiparesis, epilepsy, cerebral hemorrhage, aseptic meningitis, and fever [61]. Immune activation in the brain may cause inflammation, and excessive inflammation can trigger cerebral edema. In cases of more widespread brain edema, symptoms such as headache, altered mental status, nausea, and vomiting may occur. For example, in the clinical trial of PVSRIPO, patients developed cerebral hemorrhage and neurological symptoms [62]. We believe that the ideal state is one in which the virus can find a balance between toxicity to the tumor and safety for other tissues and maintain an optimal antitumor inflammatory response without limiting viral transmission. Some of the complications in the study were caused by an overly invasive dosing regimen, so a gentle dosing regimen is also what an ideal OV should have. A new direction in lysovirus therapy is a combination therapy, which combines the virus with other treatment strategies [63]. Combining OVs with antitumor drugs or immunostimulatory drugs can create synergistic effects and break drug resistance and immune tolerance [64,65]. OVs prevent the repair of tumor cell DNA and make the tumor more sensitive to radiation therapy [66] (Figure 2).

### 3.3. Phages Can Target Gliomas for Drug Delivery

Phages are also effective carriers for glioma therapy, have a greater safety profile, do not normally proliferate in mammalian hosts [31], and can be employed to transport medications across the BBB [67]. Phage display technology involves inserting DNA sequences of exogenous peptides into phage shell protein genes so that both peptides and shell proteins are expressed on the phage surface. Using biopanning, targeting peptides or antibodies with a high affinity for tumors are selected, and these proteins can be used as drugs or drug carriers after modification [68]. GICP (glioma-initiating cell peptide), which exhibits a strong affinity for VAV3 protein, was identified using the phage display method [69]. Thus, it is possible to deliver medications precisely to gliomas using phages. One group investigated the efficacy of systemic temozolomide-activated phage-targeted gene therapy for GBM. The single-stranded genome of the human adeno-associated virus (AAV) was inserted into M13 phage, whose capsid was designed to display an RGD4C ligand, which binds to a αvβ3 integrin receptor, which is overexpressed on tumor cells, and once they bind, RGD4C/AAVP viral particles enter the cell, upon which the AAV genome is released to express genes from the cytomegalovirus CMV promoter. The CMV promoter is replaced with the Grp78 promoter, which can be activated by GBM or stimulated by TMZ [70] (Figure 3).

## 4. Impact of the Local Environment

### 4.1. Glioma Microenvironment

In addition to glioma cells, the solid glioma tissue contains nontumor cells, such as endothelial cells, pericytes, microglia/macrophages, fibroblasts, neurons, and astrocytes, as well as soluble cytokines secreted by various cells and the extracellular matrix, which interact to form the glioma microenvironment. Nontumor cells and tumor cells can interact in the glioma microenvironment through the following five routes (Figure 4):

#### 4.1.1. NF-κB

NF-κB is a ubiquitous pleiotropic factor that regulates the expression of more than 150 genes and is intimately related with carcinogenesis, proliferation, metastasis, and treatment resistance. Multiple intracellular and extracellular stimuli, such as bacterial, viral, or interleukin, initiate the classical NF-κB signaling pathway by recruiting and activating IKK via multiple pathways. The activated IKK then induces the phosphorylation and ubiquitination of IB, which is degraded by proteases. With the aid of NLS sequences, the liberated NF-κB dimer is rapidly translocated to the nucleus through nuclear translocation and plays a role in controlling gene transcriptional expression [71]. Additionally, this system mediates the connection between tumor cells and normal cells. For instance, RANKL generated by GBM cells stimulates astrocytes via NF-κB signaling, prompting them to emit immunosuppressive regulatory factors such as TGF-β [72]. The extracellular protein fibulin-3, which is secreted by GBM cells, induces oncogenic NF-κB in tumors and enhances NF-κB activation in peritumoral astrocytes [73]. The dysregulated, constitutive activation of NF-κB signaling is a key modulator of the inflammatory and anti-immune processes associated with the advancement of cancer. In GBM, continuous NF-κB activity is required for the maintenance of tumor-initiating cell populations and the development of the tumor to a more aggressive phenotype.

#### 4.1.2. SHH

Hedgehog (Hh) ligand, Patched (Ptch), and Smoothened (Smo) are the primary proteins involved in the HH signaling pathway [74], and Hedgehog works on the Ptch receptor family. Ptch-1 and Ptch-2 impede Smo activity and prevent signaling in the absence of Hh proteins. The presence of Hh reactivates the pathway and induces downstream signaling, resulting in the activation of Gli and the regulation of transcription [75]. The dysfunction or abnormal activation of the Hh signaling system has been linked to anomalies of development and cancer. The SHH signaling pathway maintains glutamate and ATP release in astrocytes in the glioma microenvironment [76]. The SHH pathway may interact with CX43 to facilitate cell migration and drive tumor programs [77].

#### 4.1.3. P53

As a tumor suppressor transcription factor, wild-type p53 regulates the cell cycle or apoptosis in response to various stimuli, and loss of p53 function is typically a requirement for cancer development [78,79]. p53-deficient fibroblasts increase RNS production and accumulation of oxidative DNA damage products associated with the specific upregulation of endothelial nitric oxide synthase (eNOS), inducing nontumorigenic epithelial cells of oral and ovarian tissue origin to become invasive, and we hypothesize that this effect may also be present in gliomas [80]. Compared to ECM from p53(+/+) astrocytes, ECM from p53(+/−) astrocytes is rich in laminin and fibronectin and improves GBM cell survival [81].

#### 4.1.4. JAK/STAT

STAT (signal transducer and activator of transcription) is a possible cytoplasmic transcription factor that can function as an effector downstream of cytokine and a growth factor receptor signaling. Typical JAK/STAT signaling pathways involve the activation of Janus kinase (JAK) or growth factor receptor kinase, the phosphorylation of STAT proteins, their dimerization, and their translocation into the nucleus, where STAT functions as a transcription factor with pleiotropic downstream effects. STAT3 and 5, which are involved in promoting cell cycle progression, cell transformation, and preventing apoptosis, have been found in abundance in gliomas, and there is a lot of evidence that their aberrant activation contributes to glioma development [82]. There is a strong relationship between NF-κB and STAT3 in cancer, and these two transcription factors function synergistically to enhance cytokine production and cause angiogenesis and inflammatory cell infiltration [83]. Microglia can regulate STAT 3 and NF-κB activity via mTOR, promote the immunosuppressive microglia phenotype to promote glioma immune escape, and upregulate their own IL-6 secretion via TLR4 as a mitogen for glioma stem cells [84,85]. IL-6 can also bind to IL-6R to activate STAT3 via JAK and promote tumor proliferation [86]. In addition, STAT3’s interaction with hypoxia-inducible factor 1 (HIF-1) and vascular endothelial growth factor (VEGF) under local hypoxic conditions can increase VEGF expression and glioma angiogenesis [87].

#### 4.1.5. PI3K/Akt

The regulatory subunit of PI3K, p85, dimerizes and releases its catalytic subunit, p110, when PIK3 is activated. PIP2 is phosphorylated to become PIP3 by p110. PIP3 initiates recruitment of Akt to the inner membrane and recruits the downstream Akt to inner membranes and phosphorylates Akt on its serine/threonine kinasesites. Activated Akt is involved in downstream mTOR-mediated protein responses and cell cycle regulation [88]. This mechanism is intimately associated with glioma angiogenesis [89]. Through the PI3K/Akt signaling pathway [90], chronic stress also stimulates glioma cell proliferation. In recurrent GBM, macrophage-derived IGF-1 and the tumor cell IGF-1 receptor (IGF-1R) promote the PI3K pathway, and inhibiting this mechanism increases survival [91]. Additionally, this route mediates the connection between tumor cells and neurons. Neurons are activated to secrete large amounts of neuroligin into the TME. Neuroligin-3 stimulates many carcinogenic pathways, not only upregulating synapse-related genes, enabling neurons to communicate synaptically with gliomas to promote glioma growth, but also promoting GBM proliferation by activating PI3K as a mitogen [92,93,94,95].

In addition to the aforementioned pathways, there are numerous other linkages between tumor cells and nontumor cells in gliomas, so that they can influence gliomas through controlling nontumor cells, and there is also a lot of research on nontumor cells.

In addition to these cells, there are numerous additional active molecules in TME, such as cytokines, metabolic compounds, and neurotransmitters. The cytokines include IL-10, SDF-1, TGF-β, and so on. KPNA 2 is regarded as a glioma marker, and IL-10 can upregulate KPNA 2 in order to enhance the proliferation and invasion of glioma cells [96]. Neurotransmitters include glutamate and γ-Aminobutyric acid, which may be linked to glioma-induced seizures [97].

#### 4.1.6. Microorganisms in Glioma

Additionally, microorganisms are dispersed throughout the glioma microenvironment. These bacteria may be present in brain tissue prior to tumorigenesis and subsequently induce glioma development and migration, or as a result of glioma altering the local microenvironment, allow bacteria to invade the tumor from elsewhere, such as blood–brain barrier disruption and immunosuppression. Furthermore, germs may enter the brain via neurons [98]. Hu Hongrong et al. performed transcriptome and macrogenome sequencing of multiregional tumor tissue and paracancerous tissue samples from glioma patients in China, resulting in the identification of several microbial groups in gliomas, each of which was found to prefer different types of immune cells in the tumor microenvironment and may be associated with the development of gliomas. So we believe that the specific possible function of intratumoral microorganisms must be verified further.

### 4.2. The Connection between the Gut Microbiota and Glioma

As an organic whole, the various parts of the human body are intimately connected. The intestinal microbiota are the most numerous and prolific microbiome in the human body [99]. As science advances, people are becoming increasingly aware of the importance of intestinal flora to the human body and even consider them to be a component of the gastrointestinal tract. Their function is not restricted to maintaining the intestinal mucosal barrier, immunological homeostasis, and metabolic balance, but it also governs metabolic health and disorders throughout the entire human body [100]. In recent years, the microbe–gut–brain axis has become the focal point of biomedical research. This notion highlights the interplay among microorganisms, gut metabolism, and human brain nerves and presents a valuable new dynamic in the treatment of cancer and CNS illnesses. Therefore, we feel that targeting the gut–brain axis as a treatment for glioma, a tumor of the CNS, is also a promising approach. The relationship between gliomas and gut microorganisms is bidirectional. It has been demonstrated that gliomas can cause ecological dysbiosis of the intestinal microbiota in mice, as indicated by significant differences in the Firmicutes/Bacteroides (F/B) ratio, and an increase in Verrucomicrobia phylum and Akkermansia genus, and that TMZ administration can reverse the changes in microbial taxa [101]. The most significant SCFAs and neurotransmitters were reduced in human glioma patients after tumor formation compared to healthy individuals [102]. The influence of flora on gliomas is complex, and it has been demonstrated that antibiotic (ABX) administration enhances glioma growth in mice by dysregulating intestinal flora [103], although oral antibiotic treatment delayed mouse death in mice with advanced gliomas [104]. We consider that the reason for this phenomenon is that the early intestinal flora are neutral, healthy, and able to inhibit glioma progression, but as the tumor grows, the flora are regulated by it, and this creates a combination disadvantageous to the host, which instead promotes glioma progression. This phenomenon fully proves our view that glioma and intestinal flora interact with each other. We conclude that gut flora can generate compounds that influence the course of gliomas via neuronal, endocrine, and immunological pathways (Figure 5).

#### 4.2.1. Metabolites Produced by Intestinal Flora

Short-chain fatty acids (SCFAs), vitamins, polyphenols, and amino acids are among the numerous and varied metabolites produced by the human gut flora.

SCFAs are the primary metabolites produced by the fermentation of dietary fiber and resistant starch bacteria in the colon. In the periphery, SCFAs bind GPCR in gastrointestinal cells, regulate endocrine and immune function, and modulate the release of hormones and neurotransmitters, including ameliorating diabetes by increasing insulin release and reducing insulin resistance [105,106]. SCFAs can cross the BBB via MCT located on endothelial cells and by upregulating the expression of tight junction proteins [107]. In the CNS, SCFAs inhibit histone deacetylase (HDAC) activity, hence boosting the acetylation of lysine residues in nucleosomal histones found in several cell types [107,108]. Acetylation changes play a crucial regulatory function in carcinogenesis and development. SAHA, an HDAC inhibitor, has been shown to diminish the damaging effects of glioma in the tumor microenvironment [109]; therefore, we hypothesize that SCFAs may have a similar effect. SCFA supplementation corrected the immature phenotype of microglia in GF animals, and mice lacking the SCFA receptor FFAR2 mirrored the microglial cell abnormalities reported in GF circumstances [110], indicating that intestinal flora regulate CNS microglia maturation and function via SCFAs. SCFAs also can stimulate astrocytes by boosting mitochondrial activity [111]. These are tightly connected with tumor cells in the tumor microenvironment, leading to the conclusion that SCFAs regulate cancers via nontumor cells. Butyrate is a specific type of SCFA, and as mentioned previously, ROS can affect the malignancy of tumor cells via the redox-regulated transcription factor NF-κB [112]. Studies have shown that butyrate can reduce ROS by upregulating SOD1, leading us to speculate that it can inhibit tumor progression [113]. CK, a product of intestinal flora, inhibits the migration of C6 glioma cells by modulating their downstream signaling pathways [114]. The important amino acid tryptophan (Trp) is absorbed in the small intestine and enters the bloodstream. In the periphery, most of the tryptophan is converted to kynurenine (Kyn) by IDO, and a small percentage is converted to serotonin and melatonin, and it can also be metabolized directly by gut flora to indole and other compounds. Kyn and indole can flow across the BBB and act as ligands for AHR [115,116]. As discussed above, the gut microbiota play a vital role in the control of Trp metabolism. By managing intestinal flora, it is feasible to govern the equilibrium of these metabolic pathways and restrict the availability of peripheral Trp, so it is also possible to regulate the quantity of Trp that crosses the BBB [117]. The Trp entering the CNS can also be metabolized by IDO and IL4I1 to produce Kyn and indole, which will activate the AHR [118,119]. AHR has two opposite effects on glioma development. On the one hand, the Trp–Kyn–AHR pathway promotes Treg cell differentiation and also upregulates PD-1 expression levels, inhibits T cell activity, and promotes immune tolerance; on the other hand, AHR is a tumor-suppressor gene in GBM, and the deletion of AHR enhances GBM tumor growth and invasion [120]. The role of AHR has been a contentious issue, and its characteristics may be dependent on cell type and environment. The intestinal flora can metabolize and produce glutamate (Glu) [120]. As stated previously, Glu and GABA are linked to glioma-induced epilepsy [97]. Glu increases intracellular Ca^2+^ concentrations, and high levels of Ca^2+^ can support cellular events that stimulate Glu production in glioma cells, creating a positive feedback loop [121,122]. Furthermore, glutamate can drive brain tumor progression via synaptic inputs. In glioma cells, Glu is the central component of multiple important metabolic pathways [123]. The gut microbiota can both inhibit the metabolism of arginine-derived cancer-promoting metabolites by decreasing the arginine flux in vivo and create arginine-derived cancer-promoting metabolites, including polyamines and nitric oxide (NO), during assimilation of dietary arginine [124]. Extracellular polyamines induce cancer cell proliferation and migration by upregulating ornithine decarboxylase (ODC), Akt, HIF-1, and VEGF and downregulating p27 expression [125]. Increasing concentrations of NO have a dual effect on glioma: On the one hand, NO interferes with the T-cell function by inhibiting MHC class II transcription, suppressing the JAK3-STAT5 signaling pathway and inducing T-cell apoptosis; on the other hand, high levels of NO can increase mitochondrial membrane permeability, which triggers the release of cytochrome c, the production of apoptosis-inducing proteins, and the activation of specific caspase [124]. Through metabolites, gut flora can influence the growth of tumors.

#### 4.2.2. Intestinal Flora Regulate Hormone Release

There is a correlation between the existence of gut microbiota and specific alterations in hormone levels. The intestinal flora are also considered an endocrine organ because the microbiota make and secrete hormones, respond to host hormones, and modulate the expression levels of host hormones [126]. Different hormones can pass the BBB and enter the brain’s core [127]. Recent research has demonstrated the presence of numerous hormones and growth-stimulating factors in the microenvironment of GBM, which have varying effects on tumor growth [127]. According to epidemiological evidence, the incidence of GBM is higher in men than in women [128]. Earlier research demonstrated that estradiol induces JNK-dependent apoptosis in glioblastoma cells; however, a recent study found the opposite to be true. These contradictory findings highlight the role of estrogen in glioma cell growth regulation, which may depend on the type of receptor [128,129,130]. Through the secretion of glucuronidase, intestinal flora can regulate estrogen [131], which has an effect on tumors. Through the PI3K/Akt signaling pathway, stress hormones, such as glucocorticoids and norepinephrine, can promote glioma cell proliferation [90]. Insulin in neurons can prevent synaptic loss and delay death due to glioma, and intestinal flora can improve insulin resistance by producing the metabolite SCFA, and the levels of certain bacteria, such as Collinsella, are highly correlated with serum insulin levels [131,132]. Growth hormone receptor (GHR) expression is elevated in one third of patients with gliomas, and GHR signaling affects the expression of proteins involved in cell motility, enhancing cell migration, invasion, and proliferation in vitro and carcinogenesis, tumor growth, and tumor invasion in vivo [133,134]. By stimulating the PI3K/AKT signaling pathway, insulin-like growth factor-1 (IGF-1) inhibits glioma cell death [135]. In conclusion, intestinal flora can regulate the growth and apoptosis of gliomas by modulating hormone levels.

#### 4.2.3. Intestinal Flora Impact Neuronal Function

The vagus nerve (VN) is the primary component of the parasympathetic nervous system and consists of 80% afferent fibers and 20% efferent fibers. Afferent fibers of the vagus nerve are present in all layers of the gut wall, and intestinal flora stimulate intestinal endocrine cells to create hormones, such as 5-HT, cholecystokinin (CCK), and glucagon peptide, as well as their own metabolites. The VN transports these neurotransmitter-like chemicals to the brain. Therefore, the VN is a critical neuronal communication channel between the CNS and the gut microbiota and is engaged in the bidirectional connection between the gut microbiota and brain [136]. Multiple types of peripheral nerves in the tumor microenvironment have been shown to influence cancer cell behavior [137], and the vagus nerve has anti-inflammatory properties mediated through ACh and 7nAChR that have been suggested to slow tumor progression [138,139,140]. Consequently, we believe that altering VN function by changing the flora may be a promising strategy for the future treatment of glioma.

#### 4.2.4. Gut Flora Regulate the Tumor Microenvironment

Recurrence of glioma is linked to the tumor’s immunosuppressive microenvironment, and intestinal flora can modulate the activity of central immune cells. Natural killer (NK) cells are a class of cells that are directly cytotoxic to cancer cells. Abundant intestinal flora increase their surface chemokines and consequently their tumor-killing ability, while administration of ABX to mice decreases the percentage of mature NK cells [103,141]. Tumor-associated macrophages (TAMs), comprising bone marrow-derived macrophages (BMDMs) and microglia (MG) in the brain, are the most abundant cell population in the glioma TME and have been identified as a key cellular component in glioma development and malignant progression. In germ-free mice, the proportion of mature microglia decreases, while the total number of microglia remains unchanged, and the absence of microbes reduces the immune response to microglia and causes damage to microglia, which can be repaired by the microbiota or SCFA [110], suggesting that the gut microbiota regulate the maturation and function of microglia. The production of short-chain fatty acids by Bifidobacterium affects the homeostatic expansion of branching MG [142]; the production of butyrate by Clostridium butyricum attenuates MG activation and microglia-mediated neuroinflammation [143], and the ability of Lactobacillus to prenatally regulate MG malnutrition and activation indicates that different intestinal bacteria can regulate MG via distinct pathways or substances [144]. The effects of TAMs are extremely malleable and can distinguish between the M and M2 phenotypes. The M1 phenotype is considered proinflammatory and antitumor and is typically acquired after stimulation with GM-CSF, TLR4 ligand, or IFN-γ. The M2 phenotype is cytoprotective, immunosuppressive, and tumorigenic [145]. The makeup of the intestinal flora can influence the propensity for differentiation [146]. Bone MDSCs are a diverse population of bone marrow-derived cells that are progenitors of dendritic cell macrophages and granulocytes. MDSCs are able to produce NO, sequester cystine, limit the availability of cysteine and nitro LCK, secrete IL-10 and TGF-β, and upregulate PD-L1 expression. Therefore, MDSCs can inhibit T cells activation in the glioma microenvironment [147,148,149,150]. Gut flora can regulate MDSC via the GM–CSF signaling pathway and the IL-4-Stat6 pathway [151,152].

In addition to immune cells, the intestinal microbiota can affect the activity of endothelial cells through the circulation of different chemicals. Angiogenic inducers enhance endothelial cell proliferation, migration, and differentiation during neovascularization. First, proangiogenic growth factors attach to their receptors on endothelial cells (ECs), which triggers the release of matrix metalloproteinases (MMPs). These proteases degrade the basement membrane, thereby permitting the activated endothelial cells to migrate and proliferate beyond the existing vasculature. The neovascularization then spreads with the participation of adhesion molecules and MMPs. Finally, pericytes converge in the vessel wall to stabilize the newly formed vessels [153]. Intestinal flora can transcend the intestinal barrier and persist in tissues outside of the gut, resulting in persistent inflammation and disease [154]. Intestinal bacteria and their metabolites can thereby enter the circulatory system by breaching the intestinal blood barrier, and their metabolites can function as angiogenesis inducers or inhibitors. The intestinal flora-dependent metabolite trimethylamine-N-oxide, for instance, may accelerate endothelial cell senescence and vascular aging via oxidative stress [155], and the flora amino acid metabolite indolol sulfate inhibits NO synthesis and upregulates ROS, which leads to endothelial dysfunction and atherosclerosis [156]. Therefore, we hypothesize that the gut microbiota can influence glioma advancement via influencing endothelial cells and, consequently, glioma progression. It has been demonstrated that the molecular determinants of the gut microbiota, LPSs, and SCFAs regulate the survival of gut neurons, and SCFAs also stimulate neurogenesis [157]; therefore, we hypothesize that the intratumoral bacteria of glioma can influence glioma progression by regulating neurons.

Numerous studies have been conducted on the treatment of neurological diseases by modulating the gut microbiota, including the treatment of Parkinson’s disease with fecal microbiota transplantation and the use of probiotics to slow the progression of Alzheimer’s disease [158,159]. There have also been numerous studies on altering gut flora to treat cancer [160], although few have focused on gliomas. It is difficult to treat cancer by altering the gut flora. Diet, drugs, emotions, and the surrounding environment can all affect the flora. In order to construct a rational combination of microorganisms, it is necessary to clarify the effects of bacteria and metabolites on the tumor as well as the link between bacteria. Important concerns include the use of preparative regimens (e.g., antibiotics) and strategies of maintaining change (diet and prebiotic supplements) prior to modifying the gut microbiota. Brain tumor patients have a significantly lower alpha diversity index of intestinal flora compared to healthy controls and a significant dysregulation of intestinal flora structure and function, especially in malignant gliomas. Through statistical analysis, the team identified six abundant genera, namely, Fusobacterium, Akkermansia, Escherichia/Shigella, Lachnospira, Agathobacter, and Bifidobacterium [161]. This can serve as a foundation for further in-depth studies, and we anticipate early clinical application of these medicines.

## 5. Conclusions

This article discusses in detail the use of microorganisms in the treatment of glioma. In recent studies, a number of microbial derivatives have exhibited antiglioma activity, which may help in the development of new drugs. Targeted treatment of tumors by microorganisms has been widely studied in cancer, especially the use of lysoviruses in glioma, and combining lysoviral therapy with other approaches may be a direction for future research. Gut flora constitute the largest microbial community in the human body and can affect CNS tumors through the microbial–gut–brain axis, and the deficiency of gut flora can promote tumor proliferation. We have described the mechanisms by which flora may affect tumors in four respects: metabolites of gut flora, hormones, nerves, and glioma microenvironment. New therapies can be developed later by affecting these pathways. In conclusion, much further work is required, but microorganisms may offer new hope for the treatment of glioma.

## Figures and Tables

**Figure 1 cancers-14-05977-f001:**
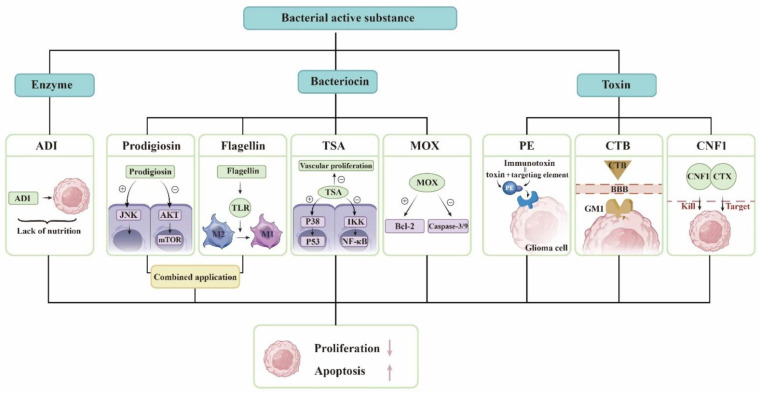
Derivatives of microorganisms are used to treat gliomas. The bacterial active substance includes enzymes, bacteriocins, and bacterial toxins. The list some of substances and their mechanisms of action. ADI exhausts arginine in the tumor and makes the tumor lack of nutrition. Prodigiosin activates the JNK pathway and reduce the AKT/mTOR pathway in GBM cells. Flagellin modulates the conversion of TAM from M2 to M1 type. TSA leads to p53 phosphorylation by activating p38MAPK and inhibits NF-κB activity by inducing IKK dephosphorylation. TSA also inhibits GBM vascular proliferation. MOX increases the ratio of Bcl-2-associated X protein/B-cell lymphoma 2 and activates the caspase-3/-9 cascade. PE and proteins that target gliomas constitute immunotoxins. CTB can pass the BBB and is targeted. CTX and CNF1 are responsible for identifying and killing glioma cells, respectively. They all promote glioma cell apoptosis and inhibit glioma proliferation.

**Figure 2 cancers-14-05977-f002:**
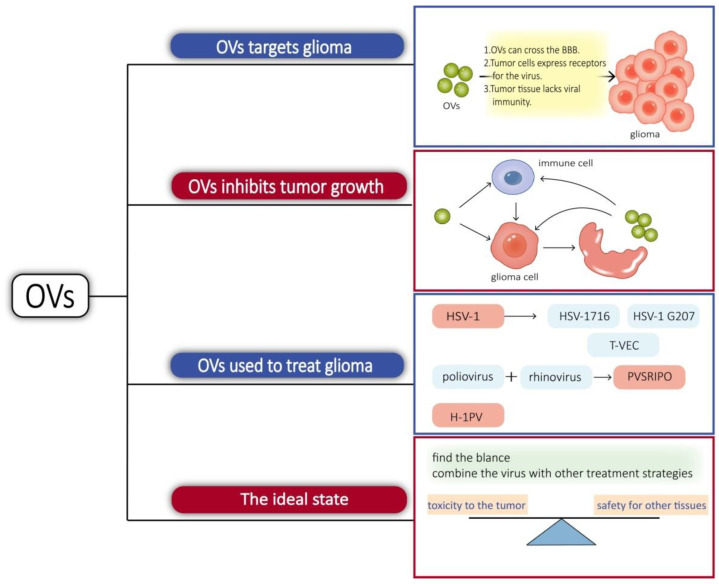
Some viruses can cross the BBB; some viruses can bind to receptors on the surface of glioma cells, and tumor tissue lacks immunity to viruses, so OVs tend to proliferate in glioma cells. On the one hand, OVs invade glioma cells, causing cells lysis and releasing new viruses that can invade other tumor cells, and, on the other hand, they can activate immune cells in tissues, breaking the immunosuppressive microenvironment of tumors and promoting immune cells to kill glioma cells. This article focuses on the research progress of three OVs: HSV-1, PVSRIPO, and H-1PV. The fastest progress is currently being made on HSV-1 research. HSV-1 is changed into HSV-1716, HSV-1 G207, and T-VEC by gene-editing technology. The ideal OVs would balance toxicity to the tumor with safety for other tissues, and another direction of development is to combine OVs with other therapies.

**Figure 3 cancers-14-05977-f003:**
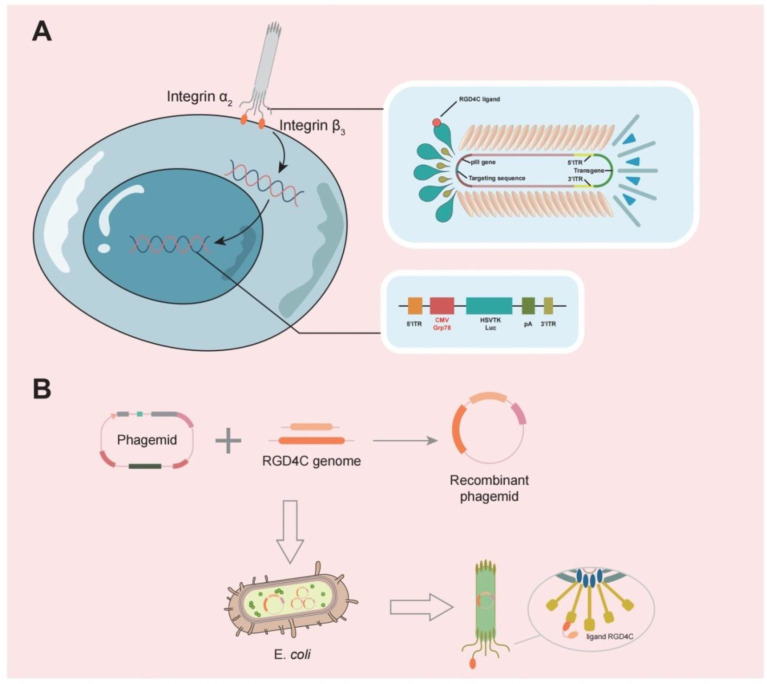
(**A**) A typical representation of M13 phage contains a single-stranded circular DNA, and the major coat proteins are pIII (green), pVIII (orange), and pVII + pIX complex (blue).The RGD4C ligand of this phage is individually designed to bind to the α2β3 integrin receptor, allowing it to bind to tumor cells and release the AAV genome into tumor cells. (**B**) The RGD4C ligand DNA sequence was inserted into the appropriate position of the phage shell protein structural gene by genetic engineering techniques, so that the RGD4C ligand gene is expressed along with the shell protein.

**Figure 4 cancers-14-05977-f004:**
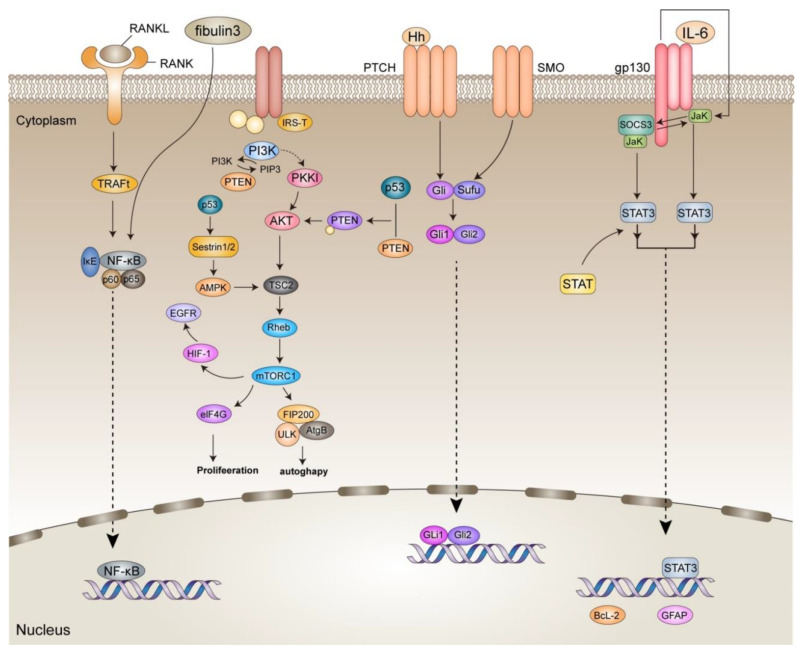
There are multiple signaling pathways in the glioma microenvironment that interact to influence the development of glioma. NF-κB is activated by upstream factors and then enters the nucleus to participate in the regulation of DNA synthesis in tumor cells. Aberrant activation of STAT has been shown to contribute to glioma proliferation, while IL-6 activates STAT3 via JAK. Akt activated by PIP3 participates in downstream mTORC1-mediated biochemical reactions. This pathway is closely related to glioma angiogenesis and also promotes glioma cell proliferation and growth.

**Figure 5 cancers-14-05977-f005:**
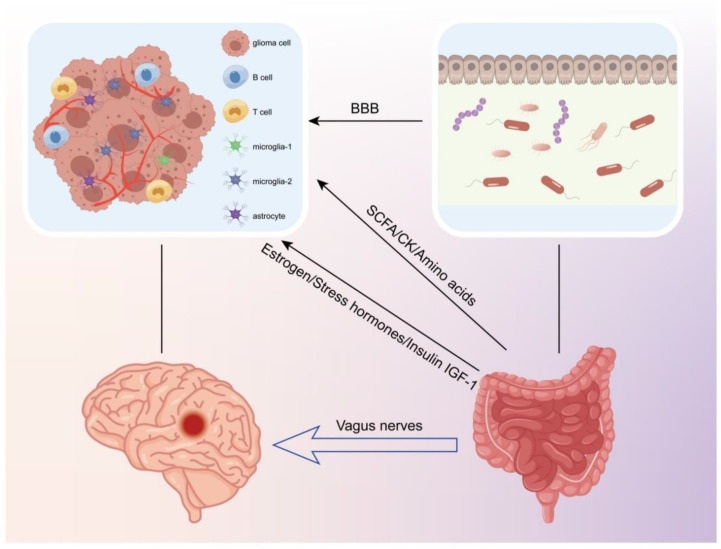
SCFAs and other metabolites produced by the gut flora can cross the blood–brain barrier(BBB) and affect glioma progression through neural, endocrine, and immune pathways. The gut flora can regulate glioma growth and apoptosis by modulating the levels of estrogen, insulin-like growth factor, and other hormones. The vagus nerve is also a key pathway for neural communication between the CNS and the gut microbiota.

**Table 1 cancers-14-05977-t001:** Exogenous microorganisms and their derivatives for glioma treatment.

Derivatives	Serratia marcescens	Prodigiosin induces autophagy and apoptosis in glioma cells [18]. Flagellin breaks immune tolerance [19].
	Pseudomonas aeruginosa	PE is used to synthesize fusion toxin to kill glioma [20].
	Streptomyces	As a deacetylase inhibitor, TSA can kill glioma cells [21]. MOX induces apoptosis in glioma cells [22].
	Marine Bacteria	Marine bacteria produce antiglioma substances [23].
	Vibrio cholerae	CTB facilitates glioma-targeted drug delivery [24].
	*E. coli.*	CNF1 affects the state of neurons and fights brain tumors [25]. Binding CNF1 to CTX forms a protein that prolongs the survival time of glioma mice [26].
	Streptococcus pyogenes	They can produce ADI [27].
Microorganisms	HSV-1	The virus is able to specifically infect tumor cells and induce tumor lysis by releasing viral progeny [28].
	Salmonella	Glioma xenografts can be targeted by injecting genetically engineered Salmonella typhimurium in the tail vein of mice [29]. Glioma mice survived longer by using genetically engineered Salmonella typhimurium that targets delivery and expresses TIMP-2 [30].
	Filamentous phage	Filamentous phages are ideal drug delivery agents [31].

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
