# Peer review of "Investigational Microbiological Therapy for Glioma"

_cancers, 2022, doi:10.3390/cancers14235977_

Round 1

Reviewer 1 Report

The authors described immunotherapy for glioma using microorganisms. The manuscript was well written. I have a minor revision.  Table 1 is so crowded and hard to read. Please change more easily for readers.  

Author Response

Thank you for recognizing our article.

We are very sorry for our negligence. We have adjusted the row spacing and column widths of table 1 and simplified the language to make the table more aesthetically pleasing and easier to read.

Special thanks to you for your good comments.

Reviewer 2 Report

I found review "Investigational Microbiological therapy for glioma" very interesting and comprehensive, talking about the field of alternative therapy for gliomas. However, the whole section 4.1. is just description of signalling pathways in glioma, with no further connection to microorganisms. Despite of that, I will recommend your article for direct publishing. 

Author Response

Thank you for recognizing our article.

It is really true as you suggested that the section 4.1. is just description of signaling pathways in glioma. But these signaling pathways are very important for the development of glioma, and in 4.2. we described the effects of gut flora on glioma, many of which are through influencing these pathways. In this revision, we removed some fragments that were less relevant to microbes in this section.

Special thanks to you for your good comments.

Reviewer 3 Report

The manuscript by Wang et al. reports various information, often -in my opinion- poorly related to the topic, about glioma, its therapy, and the use of bacteria as anticancer therapeutics.

As for the latter point, which is expected to be the topic the review should be focused on, it deals with the report of general information on bacteriotherapy and virotherapy as general anticancer strategies. However, none of them is applied specifically to brain cancers. In this regard, the literature is not abundant, but some related citations are missing (e.g., Mehta et al., 2017).

Also, par 4.1 sounds merely informative on cancer biology, but poorly related to microbial-based therapies. Similarly, the use of bacterial-derived molecules sounds out of topic.

Moreover, it is surprising that, given the unique features of brain cancers, as well as the huge problems potentially arising from the inflammatory response associated to immune activation and the presence of bacteria, the authors do not mention such limits and possible strategies to overcome them. Though bacterial therapy is considered a very promising anticancer tool in solid cancers, the unique features of the brain might strongly limit its application to these cancers. Such problems should be extensively discussed.

Moreover, I'd encourage the authors to improve the overall organization and the English language of their manuscript.

Author Response

In this article, we present investigational microbial therapies for gliomas. We focus on the role of exogenous microorganisms and their derivatives on glioma and the effect of autologous intestinal flora on glioma.

Bacterial therapies are currently widely studied in cancer but rarely in glioma applications, and in this revision we have only made a few remarks about their research potential. Viral therapies are the focus of this revision. Oncolytic viruses (OVs) have been extensively investigated in glioma. The viruses mentioned in section 3.2.2, are all used for glioma treatment, and we have focused on mentioning the most studied leading Herpes simplex virus type 1 and the Parvovirus H-1 with great potential.

It is really true as you suggested that the section 4.1. is just description of signaling pathways in glioma. But these signaling pathways are very important for the development of glioma, and in 4.2. we described the effects of gut flora on glioma, many of which are through influencing these pathways. In this revision, we removed some fragments that were less relevant to microbes in this section. The section 3.1 is ‘Derivatives of microorganisms are used to treat gliomas’. These microbial derivatives have achieved good therapeutic effects on gliomas in both animal and cellular experiments, so we consider that these substances may contribute to the development of new drugs.

Because bacterial therapy has not been extensively studied in glioma, we do not make it the focus of our discussion. We found that a review has been done by other investigators regarding the possible brain complications caused by OVs. We cite this review and briefly summarize that the key to addressing these brain complications is that the virus needs to balance its toxicity to the tumor with its safety for other tissues.

We are very sorry for our incorrect writing. Considering your suggestion, we have handed this review over to a professional for linguistic touch-ups. Please see the attachment.

Special thanks to you for your good comments.

Round 2

Reviewer 3 Report

The authors have now improved the manuscript, which is suitable for publication.